# The Role of Inflammasome-Dependent and Inflammasome-Independent NLRP3 in the Kidney

**DOI:** 10.3390/cells8111389

**Published:** 2019-11-05

**Authors:** Yang Gyun Kim, Su-Mi Kim, Ki-Pyo Kim, Sang-Ho Lee, Ju-Young Moon

**Affiliations:** 1Division of Nephrology, Department of Internal Medicine, Kyung Hee University Medical School, Seoul 02447, Korea; apple8840@hanmail.net (Y.G.K.); miya26@nate.com (S.-M.K.); lshkidney@khu.ac.kr (S.-H.L.); 2Division of Nephrology and Hypertension, Department of Internal Medicine, Inha University of Medicine, Incheon 22212, Korea; 1001happyday@hanmail.net

**Keywords:** NLRP3, inflammasome, kidney, NLRP3 inhibitor

## Abstract

Cytoplasmic nucleotide-binding oligomerization domain-like receptor protein 3 (NLRP3) forms an inflammasome with apoptosis-associated speck-like protein containing a CARD (ASC) and pro-caspase-1, which is followed by the cleavage of pro-caspase-1 to active caspase-1 and ultimately the activation of IL-1β and IL-18 and induction of pyroptosis in immune cells. NLRP3 activation in kidney diseases aggravates inflammation and subsequent fibrosis, and this effect is abrogated by genetic or pharmacologic deletion of NLRP3. Inflammasome-dependent NLRP3 mediates the progression of kidney diseases by escalating the inflammatory response in immune cells and the cross-talk between immune cells and renal nonimmune cells. However, recent studies have suggested that NLRP3 has several inflammasome-independent functions in the kidney. Inflammasome-independent NLRP3 regulates apoptosis in tubular epithelial cells by interacting with mitochondria and mediating mitochondrial reactive oxygen species production and mitophagy. This review will summarize the mechanisms by which NLRP3 functions in the kidney in both inflammasome-dependent and inflammasome-independent ways and the role of NLRP3 and NLRP3 inhibitors in kidney diseases.

## 1. Introduction

Inflammation is a defense system against pathogens in the host, and it can prevent pathogens from invading and spreading through disrupted barriers [1]. Renal inflammation occurs in response not only to pathogens but also to sterile stimuli such as ischemia, high glucose, and lipids [2,3]. Recent studies showed that sterile damage-associated molecular patterns (DAMPs) could trigger inflammation via pattern recognition receptors (PRRs), such as nucleotide-binding oligomerization domain (NOD)-like receptors (NLRs), RIG-like receptors, and Toll-like receptors (TLRs), and then induce the production of inflammatory cytokines and chemokines [4,5]. NOD-like receptor family, pyrin domain-containing-3 (NLRP3) is a cytosolic sensor and consists of a central NACHT, C-terminal LRRs, and an N-terminal pyrin domain (PYD) [6].

The NLRP3 inflammasome has been implicated in various kidney diseases, including acute kidney injury (AKI) and chronic kidney disease (CKD) [7,8,9,10]. However, blockade of IL-1β, the end product of the activated NLRP3 inflammasome, failed to mitigate kidney disease, whereupon interest concerning inflammasome-independent NLRP3 increased [11,12,13]. The process by which NLRP3 contributes to kidney injury and disease progression is assumed to encompass a combination of inflammasome-dependent and inflammasome-independent pathways [7,11].

The genetic or pharmacological inhibition of NLRP3 ameliorated renal injuries in various animal models of kidney diseases [7,8,9]. However, the detailed mechanisms by which NLRP3 is involved in the pathogenesis of kidney diseases are not yet precise. Here, we will address what has been studied so far about the role of NLRP3, primarily focusing on the mechanism of action of the inflammasome-independent and inflammasome-dependent forms of NLRP3 in kidney diseases.

## 2. The Mechanism of Action of NLRP3 in the Kidney

### 2.1. NLRP3 in the Kidney

The kidney clears not only metabolic waste products, drugs, and toxins but also bacterial toxins such as lipopolysaccharide (LPS) and cytokines from the systemic circulation. Consequently, NLRP3 could be easily activated in the kidney due to continuous exposure to various DAMPs and PAMPs (Figure 1). Renal resident mononuclear cells (RMNCs), such as macrophages and dendritic cells, contain all parts of the NLRP3 inflammasome (NLRP3, ASC, pro-caspase-1) and are able to secrete mature proinflammatory cytokines; therefore, renal RMNCs undergo caspase-1-dependent pyroptosis [8,14,15,16]. NLRP3 inducers such as LPS, silica, ATP, and radiocontrast initiate cleavage of pro-caspase-1 and release of IL-1β and IL-18 in renal RMNCs or recruit leukocytes [8,15].

In addition to RMNCs, renal tubular epithelial cells (TECs), podocytes, glomerular endothelial cells, and mesangial cells contain a substantial amount of NLRP3 [7,17,18,19]. Podocytes were identified as the primary source of IL-1β in glomerulosclerosis in both humans and mice [20,21]. Moreover, activated caspase-1 and IL-18 were increased in podocytes in response to LPS and ATP [22]. Homocysteine-induced NLRP3 inflammasome activation in murine podocytes caused cytoskeletal rearrangement in the podocytes [21]. In addition, high glucose-induced NLRP3 activation cleaved pro-IL-1β to active IL-1β in glomerular endothelial cells and podocytes [7].

Confusing results have been obtained regarding the expression of active caspase-1 and secretion of IL-1β in renal TECs. Albumin or oxidized LDL triggered the activation of the NLRP3 inflammasome and the secretion of IL-1β in TECs [23,24]. In contrast, we identified that various renal TECs, including HK-2, HKC-8, and RPTEC/TERT, and primary human renal proximal TECs did not express cleaved caspase-1 or release IL-1β under hypoxic conditions [18]. Similarly, transforming growth factor-beta (TGF-β)-stimulated renal TECs did not release pro-IL-1β and secreted very little amount of IL-18 [12]. Another study also demonstrated that LPS combined with ATP, nigericin, monosodium urate crystals, or tumor necrosis factor (TNF)α/cycloheximide (CHX) failed to induce IL-1β or IL-18 release from mouse TECs [25]. Although this topic should be researched further for confirmation, we will introduce NLRP3 in renal TECs as an inflammasome-independent protein in this review.

Renal fibroblasts express NLRP3 but do not secrete IL-1β upon LPS and ATP treatment [17]. NLRP3 in renal fibroblasts was suggested to induce tubulointerstitial fibrosis via TGF-β signaling in an inflammasome-independent manner [17]. The data from mesangial cells are incomplete and controversial. Interestingly, mouse primary mesangial cells could not release IL-1β upon LPS and ATP stimulation; however, a rat mesangial cell line (HBZY-1) secreted IL-1β under high-glucose conditions, and human mesangial cells produced IL-1β after incubation with the IgA immune complex [26,27,28,29].

### 2.2. Inflammasome-Dependent NLRP3

The activation of the NLRP3 inflammasome requires two steps: priming and activation. NF-κB activators, such as LPS, bacteria, and viruses, associate to prime the NLRP3 inflammasome [30]. Three models have been suggested to progress NLRP3 inflammasome; K^+^ efflux by extracellular ATP through purinergic 2X_7_ receptor (P2X_7_R), reactive oxygen species (ROS) generation by PAMPs and DAMPs, and lysosomal disruption of the phagocytic vesicle including crystalline [31,32,33]. Mitochondrial dysfunction was implicated as a critical factor in triggering NLRP3 inflammasome. NLRP3 agonists such as nigericin and ATP causes mitochondrial damage and reduction of NAD^+^, which promotes NLRP3 inflammasome activation [34]. Thioredoxin-interacting protein (TXNIP), as an endogenous antagonist for thioredoxin, induces NLRP3 inflammasome activation in a ROS-sensitive manner [35]. It is speculated that TXNIP, translocated to mitochondria dysfunction under oxidative stress, cause NLRP3 inflammasome activation [36].

The second step includes the oligomerization of the NLRP3 inflammasome, which mediates proteolytic cleavage of procaspase-1 into active caspase-1 to activate pro-IL-1β and pro-IL18 into IL-1β and IL-18 [37,38]. Active caspase-1 also cleaves gasdermin-D (GSDMD), which induces pyroptotic cell death [39]. Generally, activation of the NLRP3 inflammasome via caspase-1 is defined as a canonical pathway. However, a recent study identified that the NLRP3 inflammasome could secrete IL-1β and induce pyroptosis without caspase-1 and GSDMD [40]. Macrophages secreted cleaved IL-1β under LPS and CHX depending on caspase-8 during the absence of caspase-1and led to apoptotic cell death in response to nigericin [41,42]. Additionally, LPS and oxidized phospholipids directly bind to caspase 11/4/5 (caspase-11 in mice, caspase-4, and caspase-5 in humans) and lead to the activation of inflammasomes in macrophages without adaptor proteins [43,44]. This NLRP3 activation without ASC or caspase-1 is classified as the non-canonical inflammasome.

### 2.3. Inflammasome-Independent NLRP3

NLRP3 has been reported to execute various biologic functions in addition to the inflammasome, and the exact mechanisms and biologic functions of inflammasome-independent NLRP3 are being revealed. Intriguingly, NLRP3 is expressed on CD4^+^ T cells and is vital for the differentiation of type 2 T helper cells in an inflammasome-independent manner [45]. NLRP3 inhibition, but not an IL-1 receptor (IL-1R) antagonist, affected the shift of renal RMNCs to an anti-inflammatory phenotype [17].

NLRP3 was associated with TGF-β-induced epithelial-mesenchymal transition and renal fibrosis after unilateral ureteral obstruction (UUO) [12,46]. Smad phosphorylation was significantly decreased in NLRP3 KO TECs after TGF-β stimulation, and IL-1β, IL-18, and caspase-1 were not necessary for this step [12]. NLRP3 also regulated mitochondrial ROS via TGF-β1/Smad signaling in cardiac fibroblasts [13]. Although primary murine embryonic fibroblasts (pMEFs) and primary renal fibroblasts contained NLRP3, pMEFs did not release IL-1β after LPS and ATP stimulation [17]. TGF-β-induced fibrotic signals were mitigated in NLRP3-deficient fibroblasts [17]. Therefore, NLRP3 acts as an inflammasome-independent way in profibrotic changes of renal TECs and myofibroblast transdifferentiation of renal fibroblasts under injury.

NLRP3 regulates apoptotic cell death in the renal and gut epithelium as a complex with ASC and caspase-8, which is happened in mitochondria [25]. The NLRP3-ASC-caspase-8 complex was formed following stimulation with TNFα/CHX, and the apoptosis caused by TNFα/CHX was attenuated in NLRP3 KO mouse primary renal TECs [25]. Similarly, NLRP3 KO mice transplanted with wild-type (WT) bone marrow showed faster tubular epithelial repair in a murine ischemic renal injury (IRI) model [47]. Cellular apoptosis requires mitochondrial amplification for the full initiation of apoptosis [48]. Several proteins, such as cardiolipin and mitochondrial-antiviral signaling protein (MAVS), were suggested to connect and activate caspase-8 in the mitochondria during apoptosis [48,49,50]. It will be clarified whether the NLRP3-ASC-caspase-8 complex needs mitochondrial proteins. We demonstrated that renal tubular NLRP3 was located in the cytoplasm at baseline and relocated to the mitochondria under hypoxic conditions, colocalizing with MAVS [18]. Deletion of NLRP3 ameliorated mitochondrial damage and apoptosis; in contrast, overexpression of NLRP3 was sufficient to increase the level of mitochondrial ROS even under normoxic conditions [18]. Additionally, the fact that the regulation of MAVS affected mitochondrial ROS and membrane potential indicated that the interaction of NLRP3 with MAVS could be a significant activator of mitochondrial dysfunction in hypoxia [18]. Similarly, NLRP3 shRNA-treated HK-2 cells were protected from high glucose-induced ROS generation [10]. Therefore, NLRP3 regulates apoptosis by modulating mitochondrial function in response to specific stimuli apart from the inflammasome.

## 3. NLRP3 in Kidney Diseases

### 3.1. NLRP3 in AKI

#### 3.1.1. Ischemic Reperfusion Injury (IRI)

Endogenous DAMPs such as high mobility group box 1 (HMGB1), ATP, and heat-shock protein were produced with cell damage and death during IRI. These endogenous DAMPs activate the NLRP3 inflammasome in renal RMNCs and then induce tubular necrosis [51,52]. AKI caused by IRI is attenuated in NLRP3, ASC, or caspase 1 KO kidneys [11,53]. However, tubular apoptosis was ameliorated only in NLRP3 KO IRI kidneys, and WT mice engrafted with NLRP3 KO bone marrow also failed to show improvement after IRI [11]. This finding suggests that inflammasome-independent NLRP3 in nonimmune renal cells is also important in the pathogenesis of IRI. Apoptosis was also attenuated in NLRP3-deficient renal TECs during hypoxia or TNFα/CHX stimulation [18,25]. Although NLRP3 deficiency in immune cells contributes to decreasing renal IRI in part, NLRP3 inhibition in renal TECs could be the main factor in improving the initial renal tubular damage in an inflammasome-independent manner.

#### 3.1.2. Rhabdomyolysis-Induced AKI (RIAKI)

The catalytic iron or myoglobin heme from damaged muscle induced lipid peroxidation and oxidative stress in the renal TECs and led to inflammation via NF-κB signaling, which is essential for NLRP3 inflammasome activation in rhabdomyolysis [54,55]. Renal tubular apoptosis and inflammation preceded renal leukocytes infiltration at 24 h after glycerol-induced rhabdomyolysis [56]. Tubular injuries were clearly attenuated in NLRP3, ASC, caspase-1, and IL-1β KO mice during the early stage of rhabdomyolysis [56]. However, a caspase-1 inhibitor did not decrease tubular cell apoptosis despite the prevention of renal dysfunction and immune cell infiltration [57]. The discrepancy of those results suggests that the genetic inhibition of caspase-1 has another effect beyond inhibiting caspase-1. Alternatively, caspase-1 inhibition itself might have a partial ability to reduce tubular cell apoptosis in RIAKI. There have been no data indicating how NLRP3 can modulate renal TECs or RMNCs upon myoglobin stimulation. Taken together, these results suggest that NLRP3 acts as the main regulator of renal injury by regulating early tubular apoptosis in RIAKI.

#### 3.1.3. Contrast-Induced AKI (CI-AKI)

Contrast induces vasoconstriction by blocking nitric oxide-induced vasodilation, the elevation of intracellular calcium, and adenosine [58]. Furthermore, sustained vasoconstriction leads to ROS formation, medullary ischemia, and tubular necrosis. Several studies have suggested that pyroptosis or apoptosis in TECs is the main factor that induces CI-AKI [59,60]. A recent study demonstrated that contrast administration led to NLRP3 inflammasome activation in renal macrophages and CI-AKI was ameliorated in NLRP3 KO mice [16]. Urinary caspase-1 and IL-18 were detected in patients receiving coronary angiography 24 h after the procedure. In addition, inflammasome-independent NLRP3 affected CI-AKI insignificantly since contrast-induced cell death was not lessened in an in vitro study using NLRP3 KO renal TECs. Therefore, inflammasome-dependent NLRP3 in renal macrophages was speculated to have a pivotal function in promoting CI-AKI.

### 3.2. NLRP3 in CKD

#### 3.2.1. Diabetic Nephropathy

Hyperglycemia and its related metabolic rearrangement could act as DAMPs that are detected by NLR receptors [61,62]. The transcripts of NLRP3, ASC, and proinflammatory cytokines increased in monocytes from patients with type 2 DM [63]. Intriguingly, an increase in plasma levels of IL-1β and IL-18 occurred prior to the presentation of albuminuria and renal histologic changes in murine diabetic nephropathy [7]. Therefore, systemic activation of the NLRP3 inflammasome might play an essential role in the development and progression of diabetic nephropathy. NLRP3 KO mice demonstrated repression of diabetic nephropathy in type 1 and type 2 diabetes by blocking NLRP3-mediated mitochondrial ROS generation [7,10]. Inhibition of TXNIP abrogated diabetic nephropathy through inhibition of NLRP3 inflammasome activation in murine streptozotocin (STZ)-induced diabetes [64,65]. However, diabetic nephropathy was improved in WT bone marrow transplanted-NLRP3 KO mice but not in NLRP3 KO bone marrow transplanted-WT mice [7]. This suggested that NLRP3 in nonimmune renal cells could be critical for aggravating diabetic nephropathy. Podocytes and glomerular endothelial cells contain NLRP3 and release cleaved IL-1β under high-glucose stimulation. Additionally, high glucose led to NLRP3 inflammasome activation in mesangial cells, and NLRP3 inhibition decreased IL-1β release from mesangial cells [19,29,65]. NLRP3-silenced renal TECs were also protected from high glucose-induced ROS injury [10]. We demonstrated that uric acid-stimulated renal TECs recruited macrophages via C-X-C motif chemokine 12 and HMGB1, and then the macrophages produced IL-1β, which again resulted in NF-κB activation in TECs [66]. The cross-talk between macrophages and renal TECs contributed to boosting inflammation in the kidney in OLETF rats [66]. Overall, NLRP3 plays a pivotal role in diabetic kidney disease progression in both inflammasome-dependent and inflammasome-independent ways, depending on the type of renal cells.

#### 3.2.2. High-Fat Diet-Induced Renal Fibrosis

Dysregulation of endoplasmic reticulum stress and elevated fatty acids in obesity activate inflammatory signaling and result in insulin resistance [67]. Ceramide activates the NLRP3 inflammasome in bone marrow-derived macrophages and adipose tissue [68]. In addition, the NLRP3 inflammasome decreases lipid breakdown by blocking the sirtuin-1/AMP-activated protein kinase pathway and thus aggravates lipid accumulation and metainflammation in renal TECs [24]. Compared with HFD-fed WT mice, HFD-fed NLRP3 KO mice demonstrated better insulin sensitivity and lipid profiles, attenuated inflammatory cytokines in adipose tissue, and decreased renal inflammation and fibrosis [35,68,69,70,71]. P2X_7_R depletion protected mice from HFD-induced renal inflammation and fibrosis [22]. The authors showed that NLRP3 inflammasome activation was attenuated in P2X_7_R-silenced podocytes under LPS with ATP stimulation.

However, prolonged lipid overload led to lysosomal dysfunction and ultimately the depression of autophagic flux [72]. The heavy condensation of swollen and fragmented mitochondria in autophagy-deficient mice induced activation of the NLRP3 inflammasome in the kidney [72]. We ascertained the possibility of upregulated autophagic activity even in unstimulated conditions in NLRP3-depleted renal TECs [18]. Further studies are required to determine the role of NLRP3 in lipotoxicity-induced autophagy/lipophagy.

#### 3.2.3. Obstructive Nephropathy

UUO is a model of CKD caused by progressive tubulointerstitial fibrosis. NLRP3 KO mice presented reductions in tubular apoptosis, inflammation, and fibrosis following UUO [18,46,73]. Leukocytes recruitment to the kidney was dampened along with decreased inflammatory cytokines in the NLRP3 KO kidney at 14 days after UUO [46]. In accordance with animal studies, NLRP3 mRNA was increased in human fibrotic renal diseases and positively correlated with serum creatinine levels [46]. Injection of a genetically modified bone marrow-derived vehicle cells transduced with IL-1R antagonist ameliorated macrophages infiltration and interstitial fibrosis in UUO [74]. However, hematopoietic NLRP3 KO did not reverse UUO-induced renal fibrosis [46]. We identified that NLRP3 deficiency preserved obstruction-induced mitochondrial damage and tubular apoptosis in UUO mice and increased mitophagy in NLRP3 KO TECs and the kidney [18]. The upregulated mitophagy in NLRP3-deficient kidneys could be one of the critical factors for protecting cells against mitochondrial dysfunction [18]. Therefore, the effects of inflammasome-independent NLRP3 as a mitochondrial regulator in renal TECs should be elucidated.

#### 3.2.4. Crystalline Nephropathy

Crystals activate NLRP3 via potassium flux induced by a lysosomal rupture in phagocytic myeloid cells [75]. Oxalate and adenine crystal accumulation in the kidney activates the inflammasome, which is the main pathogenesis for the progression of crystalline nephropathy [14,15,76]. Crystalline-induced renal injury and fibrosis were ameliorated in NLRP3- or ASC-deficient mice [8,17]. Nevertheless, an IL-1R inhibitor did not show a beneficial effect on the development of nephrocalcinosis and the progression to CKD in crystalline nephropathy [17]. However, the pharmacologic inhibition of NLRP3 using MCC950 and β-hydroxybutyrate rescued the kidney from adenine- and calcium oxalate-induced inflammation and fibrosis even though crystal deposition was similar to that in untreated mice [8,17]. NLRP3 in renal fibroblasts promoted fibrosis by augmenting TGF-β and Smad signaling without generating IL-1β, and thus NLRP3 inhibition could dampen nephrocalcinosis [17]. Intriguingly, NLRP3 inhibition with β-hydroxybutyrate induced the polarization of intrarenal macrophages and led to a more anti-inflammatory M2c-like phenotype. Further studies are needed to understand the role of NLRP3 in the phenotype of crystal-induced renal RMNCs and whether this phenotype is independent of systemic immune cells.

#### 3.2.5. Lupus Nephritis

Immune complexes with DNA or RNA antigens directly stimulate the NLRP3 inflammasome or trigger inflammasome activation through TLR-dependent NF-κB signaling in systemic lupus erythematosus [77,78]. The kidneys of lupus-prone NZBWF1 mice demonstrated the activation of the NLRP3 inflammasome, and the renal expression of NLRP3 was correlated with the progression of CKD in patients with lupus nephritis [46,79]. A gain-of-function mutation of NLRP3 in pristine-induced lupus mice exhibited high mortality, high levels of proteinuria, and severe pathologic changes in the kidney [80]. Pharmacologic inhibition of the NLRP3 inflammasome using P_2_X7 or NF-κB blockade rescued lupus nephritis in lupus-prone animals [81,82]. Lupus-induced renal injuries were ameliorated in mice with myeloid cell-specific deletion of NLRP3 [80]. A recent study reported the upregulation of the NLRP3 inflammasome in podocytes in renal biopsies from patients with lupus nephritis [83]. Podocytes from lupus-prone NZM3329 mice exhibited an elevation of cleaved caspase-1, which was decreased by MCC950 treatment [83]. In the kidneys of lupus-prone MRL-Fas^lpr^ mice, IL-18 was not increased in infiltrating monocytes but rather in renal TECs [84]. However, it is under debate whether tubular IL-18 originated from an inflammasome-dependent way [85,86]. Taken together, these findings indicate that inflammasome-dependent NLRP3 in immune cells and podocytes is important for lupus nephritis progression.

#### 3.2.6. IgA Nephropathy (IgAN)

IgAN is progressive glomerulonephritis involving mesangial deposition of immune complexes with galactose-deficient IgA1, leading to upregulated innate immunity and activation of the complement cascade [87]. IgA-containing immune complexes activate the NLRP3 inflammasome in macrophages [88]. The genetic loss of NLRP3 rescued renal injury in a murine IgA model via inhibition of immune complex-mediated immune cell activation [88]. An IL-1R antagonist reduced proteinuria and improved renal function and pathologic changes in a murine IgAN model [89]. A human study reported an increase in NLRP3 mRNA in the serum of patients with IgAN, and NLRP3 mRNA was augmented in renal biopsies from patients with IgAN compared with those from healthy controls [46,90]. Curiously, low NLRP3 mRNA levels in renal biopsies were correlated with a high risk of end-stage renal disease and a twofold increase in serum creatinine [90]. Considering NLRP3 locates mainly tubular cells, NLRP3 mRNA expression could be decreased according to tubular atrophy and interstitial fibrosis progression. In summary, NLRP3 is implicated in the pathogenesis of IgAN in an inflammasome-dependent manner and tubular NLRP3 is also related to IgAN progression.

#### 3.2.7. Hypertensive Nephropathy

Recent evidence reported that low-grade chronic inflammation contributes to the pathogenesis of hypertension [91,92]. Several studies have implicated NLRP3 inflammasome activation in the kidneys of salt-sensitive hypertension [93,94]. Notably, hypertension was not induced in NLRP3 KO angiotensin II-infused pregnant mice, and it was not observed in ASC KO mice [95]. Noncoding mutations in NLRP3 were reported to be linked to hypertension susceptibility, and the expression of IL-1β was elevated in patients with essential hypertension [96,97,98].

NLRP3 inhibition decreased blood pressure, renal inflammation, renal immune cell recruitment, and fibrosis in salt-sensitive mice [93]. NLRP3 deficiency protected the kidney from mitochondrial dysfunction in angiotensin II-infused mice and siNLRP3-treated podocytes ameliorated angiotensin II-induced mitochondrial dysfunction and loss of podocin and nephrin [99]. Aldosterone administration leads to NLRP3 inflammasome activation in not only macrophages but also podocytes, and eplerenone and antioxidant suppressed the activation [100,101]. NLRP3-deficient mice exhibited improvements in albuminuria and podocyte injuries in response to aldosterone injection [101]. However, tubulointerstitial fibrosis was not completely improved in WT mice engrafted with ASC KO bone marrow in aldosterone-infused renal disease [100]. IL-1R antagonists also demonstrated minimal effects on renal inflammation and immune cell infiltration despite antihypertensive action [102]. These findings suggest that an inflammasome-independent mechanism could be relevant and such studies are still rare. Nevertheless, NLRP3 is suggested to be a critical component among the multiple factors that induce hypertension.

## 4. Inhibitors of the NLRP3 Inflammasome in Kidney Diseases

Several medications targeting the NLRP3 inflammasome have been developed with various binding sites [103]. Initially, medications targeting IL-1β were developed to block IL-1β as the final product of the NLRP3 inflammasome. A monoclonal antibody against IL-1β (canakinumab) and a recombinant human IL-1R antagonist (anakinra) have been used as examples [104,105]. However, several NLRP3-related diseases were not controlled by using these IL-1β inhibitors, suggesting that inflammasome-independent NLRP3 plays a role in these diseases [11,17]. We summarized the NLRP3 inhibitors whose effects were determined by renal diseases in Table 1.

Tranilast was approved initially to attenuate inflammation in allergic diseases by regulating histamine release from mast cells [111]. A recent study showed that tranilast inhibited the activation of caspase-1 and IL-1β release in macrophages in a dose-dependent manner [79]. The inhibitory effect on inflammasomes is selective for NLRP3, not NLRC4 or AIM2 [112]. Tranilast prevents NLRP3 assembly through inhibition of NLRP3-NLRP3 and NLRP3-ASC interactions. Tranilast attenuated tubulointerstitial fibrosis in UUO kidneys and CKD-associated peritoneal fibrosis [106,107]. Furthermore, it showed dose-dependent improvement of diabetic tubulointerstitial fibrosis in STZ-induced diabetic kidneys despite an absence of changes in glucose or proteinuria [108]. A reduction in TGF-β by mast cell-derived inflammatory cytokines might be the major factor in ameliorating organ fibrosis. Tranilast prevented HFD-induced insulin resistance, hyperglycemia, and IL-1β elevation in the serum, liver, and adipose tissue. However, it was also effective in decreasing IL-6 and TNF production, and thus, a beneficial effect of blocking NF-κB signaling could not be excluded [113].

Bay 11-7082 prevents the production of IL-1β and IL-18 and inhibits activation of the NLRP3 inflammasome in an NF-κB-independent manner in macrophages [109]. Bay 11-7082 decreased caspase-1 activation under ATP, nigericin, uric acid stimulation, and pyroptotic cell death following ATP in macrophages [114]. Bay 11-7082 directly inhibited NLRP3-dependent hydrolysis of ATP and had a partial effect on the activation of NLRP1 and NLRC4 inflammasomes [115]. Intraperitoneal BAY 11-7082 ameliorated HFD and high-fructose diet-induced insulin resistance, hyperlipidemia, and fatty liver [109]. Moreover, it had a beneficial effect on decreasing NLRP3 inflammasome expression and profibrogenic markers in the kidney and liver. The effect suggested a reduction in HFD-induced inflammation and blockade of subsequent renal fibrosis via TGF-β/Smad signaling [109]. BAY 11-7082 improved proteinuria, blood urea nitrogen, and renal inflammation, and ultimately, it decreased mortality in lupus-prone MRL/*lpr* mice by attenuating NLRP3 inflammasome activation and NF-κB [82]. In STZ-induced diabetic rats, oxidant stress and inflammatory cytokine levels in diabetic nephropathy were attenuated by BAY 11-7082 [110]. However, the beneficial effect of Bay 11-7082 can originate from blocking NF-κB signaling or the NLRP3 inflammasome or both.

β-Hydroxybutyrate (BHB) is produced in the mammalian liver and acts as an alternative source of ATP during energy deprivation or a low-carbohydrate diet [116]. Current evidence has reported that BHB inhibited activation of the NLRP3 inflammasome by blocking K^+^ efflux and inhibiting ASC oligomerization [115]. BHB did not inhibit NLRC4, AIM2 inflammasomes, or noncanonical NLRP3 inflammasome activation [115]. Moreover, BHB attenuated IL-1β release and caspase-1 activation in Muckle-Wells Syndrome and monosodium urate-induced gout and peritonitis animal models [115,117]. A recent study demonstrated that BHB contributed to delaying renal fibrosis in crystalline nephropathy by reducing profibrotic intrarenal macrophages and the expression of TGF-β/Smad3 in renal fibroblasts [17]. Notably, dietary restriction or pharmacologic approaches to increase BHB could be a promising therapy for attenuating NLRP3-related inflammatory diseases.

MCC950 is a small-molecule inhibitor of NLRP3 [118], and it blocks both inflammasome-dependent and inflammasome-independent NLRP3 [118]. A recent study demonstrated that MCC950 inhibited to hydrolyze ATP and blocked NLRP3 activation via direct interaction with the Walker B motif in the NLRP3 NATCH domain [119]. Notably, it did not show blocking effects on NLRP1, NLRC4, or AIM2 inflammasomes [118]. MCC950 decreased proteinuria and histologic changes related to podocyte foot process effacement in lupus nephritis mice [83]. MCC950-treated hypertensive mice showed attenuation of renal interstitial collagen deposition, inflammatory cytokines, and immune cell infiltration, such as M2-like macrophages and IFN-γ-producing T cells [93]. Increased blood pressure and albuminuria were partially reversed by MCC950, suggesting that it has a beneficial effect on volume control by alternating sodium handling [93]. MCC950 prevented renal fibrosis in crystalline nephropathy by suppressing the inflammasome in renal dendritic cells [8]. Intraperitoneal MCC950 treatment in *db/db* mice decreased albuminuria, serum creatinine, and pathologic changes, and improved renal cortical fibrosis [29]. However, few studies have shown an effect of MCC950 on NLRP3 in renal parenchymal cells. Further investigations are needed to elucidate the effect of NLRP3 inhibitors on nonimmune renal cells in NLRP3-related kidney diseases.

## 5. Summary and Future Directions

Activation of inflammasome-dependent NLRP3 is important for renal RMNCs and parenchymal cells in relation to sterile and chronic inflammation in various kidney diseases. Additionally, inflammasome-independent NLRP3 plays a pivotal role in kidney disease by regulating apoptosis, fibrosis, and mitochondrial injury in cases of direct injury to renal TECs and fibroblasts. This distinct role of NLRP3 in the kidney could be clarified with conditional, cell-type-specific NLRP3 gene modulation. To apply NLRP3 as a potential treatment target in kidney disease, it is essential to understand the exact mechanisms of action of NLRP3 beyond secreting IL-1β and IL-18. The effect of β-hydroxybutyrate or calorie restriction might be associated with regulating NLRP3 in an inflammasome-independent manner by increasing autophagy and attenuating mitochondrial dysfunction. To elucidate these effects, alterations in cell biology related to NLRP3 should be investigated further as a regulator of autophagy and mitochondrial dynamics.

## Figures and Tables

**Figure 1 cells-08-01389-f001:**
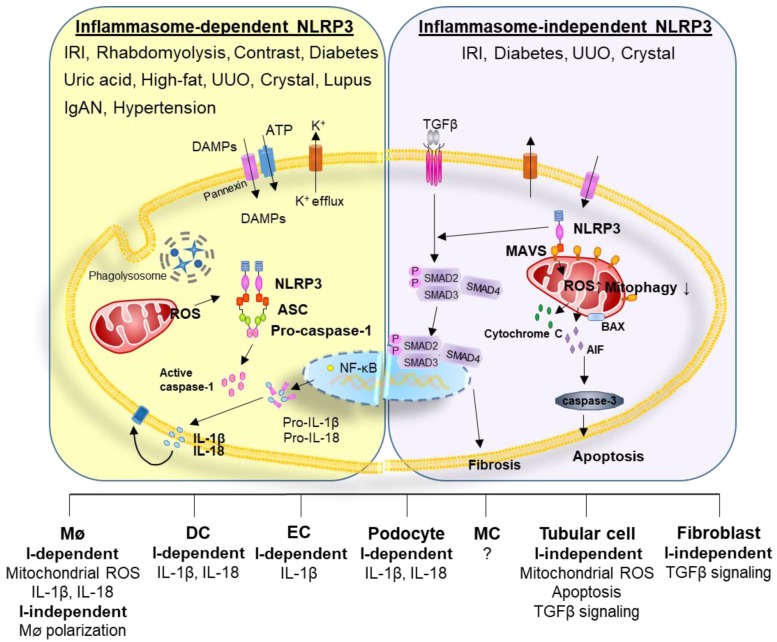
NLRP3 in the kidney operates in an inflammasome-dependent or inflammasome-independent manner, depending on the DAMPs and stimulated cells involved. The NLRP3 inflammasome is activated in various renal diseases in renal RMNCs and non-immune cells. Additionally, NLRP3 could regulate macrophage polarization via an inflammasome-independent way. NLRP3 leads to the generation of mitochondrial ROS and tubulointerstitial fibrosis in renal TECs. NLRP3 in renal fibroblasts is implicated in mediating renal fibrosis. AIF = apoptosis-inducing factor, BAX = bcl-2-like protein 4, DAMPs = damage-associated molecular patterns, DC = dendritic cell, EC = endothelial cell, IgAN = IgA nephropathy, IRI = ischemic reperfusion injury, I-dependent = inflammasome-dependent, I-independent = inflammasome-independent, MΦ = macrophage, MC = mesangial cell, NF-κB = nuclear factor kappa-light-chain-enhancer of activated B cells, ROS = reactive oxygen species, UUO = unilateral ureteral obstruction.

**Table 1 cells-08-01389-t001:** The mechanisms and related renal disease models of NLRP3 inhibitors.

NLRP3 Inhibitors	Mechanisms	Renal Disease Models
Tranilast	Inhibition of NLRP3-NLRP3	UUO [106]
	and NLRP3-ASC interactions	5/6 nephrectomy [107]
		Diabetic nephropathy [108]
BAY 11-7082	Inhibition of NLRP3 ATPase	HFD nephropathy [109]
		Lupus nephritis [82]
		Diabetic nephropathy [110]
MCC950	Blockade of ATP hydrolysis	Salt-sensitive hypertension [93]
	in NLRP3 NACHT domain	Crystal nephropathy [8]
		Lupus nephritis [83]
		Diabetic nephropathy [29]
BHB	Inhibition of ASC oligomerization	Crystal nephropathy [17]

BHB = β-Hydroxybutyrate, HFD = high-fat diet, UUO = unilateral ureteral obstruction.

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
