# Peer review of "The Role of Inflammasome-Dependent and Inflammasome-Independent NLRP3 in the Kidney"

_cells, 2019, doi:10.3390/cells8111389_

Round 1

Reviewer 1 Report

The review by Kim et al. attempts to provide a model for inflammasome-dependent and inflammasome-independent NLRP3 mechanisms in the kidney.  However, many of the statements and arguments made in this review are not referenced properly.  For example, lines 83-99 do not have a single reference listed. In addition, the evidence for an inflammasome-independent role of NLRP3 is based upon studies not carried out in kidney cells and therefore are extrapolated to the make categorical conclusions to support the author’s model.  Table 1 summarizes the mechanisms and related experimental studies on NLRP3 inhibitors.  However, most studies listed in this Table do not relate to kidney disease and therefore do not provide much relevance to the review.   The authors are encouraged to edit the manuscript, remove the non-relevant papers that do not support the role of NLRP3 in kidney disease and properly reference the relevant literature.

Author Response

Reviewer 1

The review by Kim et al. attempts to provide a model for inflammasome-dependent and inflammasome-independent NLRP3 mechanisms in the kidney. However, many of the statements and arguments made in this review are not referenced properly. For example, lines 83-99 do not have a single reference listed. In addition, the evidence for an inflammasome-independent role of NLRP3 is based upon studies not carried out in kidney cells and therefore are extrapolated to the make categorical conclusions to support the author’s model. Table 1 summarizes the mechanisms and related experimental studies on NLRP3 inhibitors. However, most studies listed in this Table do not relate to kidney disease and therefore do not provide much relevance to the review.   The authors are encouraged to edit the manuscript, remove the non-relevant papers that do not support the role of NLRP3 in kidney disease and properly reference the relevant literature.

: We appreciate your comments.

1) The references were not listed since the content in lines 83-99 was written as a figure legend. The figure legend are summarized breifly (revised line 94-102).

à NLRP3 in the kidney operates in an inflammasome-dependent or inflammasome-independent manner, depending on the DAMPs and stimulated cells involved. The NLRP3 inflammasome is activated in various renal diseases in renal resident mononuclear cells (RMNCs) and renal resident cells. Additionally, NLRP3 could regulate macrophage polarization via an inflammasome-independent way. NLRP3 leads to the generation of mitochondrial ROS and tubulointerstitial fibrosis in renal TECs. NLRP3 in renal fibroblasts is implicated in mediating renal fibrosis. AIF = apoptosis-inducing factor, BAX = bcl-2-like protein 4, DAMPs = damage-associated molecular patterns, DC = dendritic cell, EC = endothelial cell, IgAN = IgA nephropathy, IRI = ischemic reperfusion injury, I-dependent = inflammasome-dependent, independent = inflammasome-independent, MФ = macrophage, MC = mesangial cell, NF-κB = nuclear factor kappa-light-chain-enhancer of activated B cells, ROS = reactive oxygen species, UUO = unilateral ureteral obstruction

2) As your recommendation, we try to handle the content regarding experiments using kidney cells.

We add the experiments using renal resident mononuclear cells (RMNCs) including renal macrophages and dentritic cells (Kidney Int 2016;90:525-549, JCI 2013;123:236-246) (revised line 67-69).  

à NLRP3 inducers such as LPS, silica, ATP, and radiocontrast initiate cleavage of pro-caspase-1 and release of IL-1β and IL-18 in renal RMNCs or recruit leukocytes [8, 15].

NLRP3 related data studied in nonimmune renal cells was summarized (revised line 70-91).

à In addition to RMNCs, renal TECs, podocytes, glomerular endothelial cells, and mesangial cells contain a substantial amount of NLRP3[7,17-19]. Podocytes were identified as the primary source of IL-1β in glomerulosclerosis in both humans and mice [20,21]. Moreover, activated caspase-1 and IL-18 were increased in podocytes in response to LPS and ATP [22]. Homocysteine-induced NLRP3 inflammasome activation in murine podocytes caused cytoskeletal rearrangement in the podocytes [21]. In addition, high glucose-induced NLRP3 activation cleaved pro-IL-1β to active IL-1β in glomerular endothelial cells and podocytes [7].

Confusing results have been obtained regarding the expression of active caspase-1 and secretion of IL-1β in renal tubular epithelial cells (TECs). Albumin or oxidized LDL triggered the activation of the NLRP3 inflammasome and the secretion of IL-1β in TECs [23,24]. In contrast, we identified that various renal TECs, including HK-2, HKC-8, and RPTEC/TERT, and primary human renal proximal TECs did not express cleaved caspase-1 or release IL-1β under hypoxic conditions [18]. Similarly, transforming growth factor-beta (TGF-β)-stimulated renal TECs showed pro-IL-1β and very little IL-18 [12]. Another study also demonstrated that LPS combined with ATP, nigericin, monosodium urate crystals, or tumor necrosis factor (TNF)α/cycloheximide (CHX) failed to induce IL-1β or IL-18 release from mouse TECs [25]. Although this topic should be researched further for confirmation, we will introduce NLRP3 in renal TECs as an inflammasome-independent protein in this review.

Renal fibroblasts express NLRP3 but do not secrete IL-1β upon LPS and ATP treatment [17]. NLRP3 in renal fibroblasts was suggested to induce tubulointerstitial fibrosis via TGF-β signaling in an inflammasome-independent manner [17]. The data from mesangial cells are incomplete and controversial. Interestingly, mouse primary mesangial cells could not release IL-1β upon LPS and ATP stimulation; however, a rat mesangial cell line (HBZY-1) secreted IL-1β under high-glucose conditions, and human mesangial cells produced IL-1β after incubation with the IgA immune complex [26-29].

3) We agree with the comments regarding to the NLRP3 inhibitors. Table 1 is also modified so that only NLRP3 inhibitors evaluated in kidney diseases are cleaned up.

Table 1. The mechanisms and related renal disease models of NLRP3 inhibitors

NLRP3 inhibitors

Mechanisms

Renal disease models

Tranilast

Inhibition of NLRP3-NLRP3

UUO [106]

and NLRP3-ASC interactions

5/6 nephrectomy [107]

Diabetic nephropathy [108]

BAY 11-7082

Inhibition of NLRP3 ATPase

HFD nephropathy [109]

Lupus nephritis [82]

Diabetic nephropathy [110]

MCC950

Blockade of ATP hydrolysis

Salt-sensitive hypertension [93]

in NLRP3 NACHT domain

Crystal nephropathy [8]

Lupus nephritis [83]

Diabetic nephropathy [29]

BHB

Inhibition of ASC oligomerization

Crystal nephropathy [17]

BHB = β-Hydroxybutyrate, HFD = high-fat diet, UUO = unilateral ureteral obstruction

4) Many other changes have been made. In particular, the paragraphs of ‘contrast nephropathy’ and ‘hypertensive nephropathy’ were added to the ‘III. NLRP3 in kidney diseases’ section (revised line 182-191, 285-303).        

Reviewer 2 Report

The review of Kim et al. is summarizing the mechanism of NLRP3 inflammasomes in the kidney. The focus of this review is on both the inflammasome-dependent and inflammasome-independent mechanisms as well as the role of NLRP3 and NLRP3 inhibitors in kidney diseases.

The topic of this manuscript is interesting. However, I have some major concerns and suggestions.

Major comments:

The authors state in the introduction ”For inflammasomeactivation, NLRP3 must form a multiprotein complex with ASC and the cysteine protease caspase-1, which involves cleaving pro-IL-1β and pro-IL-18 to IL-1β and IL-18”. In leukocytes it is known that NLRP3 inflammasome activation and IL-1beta release typically requires two consecutive danger signals e.g. LPS and ATP (see Gross, O., C. J. Thomas, G. Guarda, and J. Tschopp. 2011. The inflammasome: an integrated view. Immunol. Rev. 243: 136–151.; Rathinam, V. A., S. K. Vanaja, and K. A. Fitzgerald. 2012. Regulation of inflammasome signaling. Nat. Immunol. 13: 333–342.). How about NLRP3 in the kidney? Furthermore, the review is mainly focusing on caspase-1. However, there is increasing evidence that other caspases e.g. caspase-8 and -11 are important and can induce non-canonical NLPR3 inflammasome activation in immune cells. Is there anything known in the kidney?

Figure 1: the figure legend is incomplete and difficult to follow. Please define the abbreviations e.g. IRI, LPS, ROS, PAMPs … and bring the terms in an alphabetic order.

Multiple references are missing: Line 64, 73, 107,111, 115, 130-134, 151, 202, 229, 243, 254, 256, 269, 270, 297, 307, 311, 318.

Line 186-187: From my point of view this is an overstatement. Please rephrase the sentence.

In general, it is sometimes difficult for the reader to understand what models/cells/tissues species (human or mouse) is discussed for example see 192- 194.

Line 238-250: How about the inflammasome-independent mechanisms?

Minor comments:

Line 10-12: Please rephrase the sentence and include caspase-1 activation and cleavage of pro-IL-1beta and IL-18.

Line 14-15: Please rephrase the sentence.

Line 20: ….in the kidney in both inflammasome-dependent….

Line 22: Please revise the Keywords.

Line 52: a blank is missing.

Line 56: define LPS

Line 164: are produced

Line 243: imply

Line 282: in a dose-dependent

Line 316: However, it…..

Author Response

Reviewer 2

The review of Kim et al. is summarizing the mechanism of NLRP3 inflammasomes in the kidney. The focus of this review is on both the inflammasome-dependent and inflammasome-independent mechanisms as well as the role of NLRP3 and NLRP3 inhibitors in kidney diseases.

The topic of this manuscript is interesting. However, I have some major concerns and suggestions.

Major comments:

The authors state in the introduction ”For inflammasome activation, NLRP3 must form a multiprotein complex with ASC and the cysteine protease caspase-1, which involves cleaving pro-IL-1β and pro-IL-18 to IL-1β and IL-18”. In leukocytes it is known that NLRP3 inflammasome activation and IL-1beta release typically requires two consecutive danger signals e.g. LPS and ATP (see Gross, O., C. J. Thomas, G. Guarda, and J. Tschopp. 2011. The inflammasome: an integrated view. Immunol. Rev. 243: 136–151.; Rathinam, V. A., S. K. Vanaja, and K. A. Fitzgerald. 2012. Regulation of inflammasome signaling. Nat. Immunol. 13: 333–342.). How about NLRP3 in the kidney? Furthermore, the review is mainly focusing on caspase-1. However, there is increasing evidence that other caspases e.g. caspase-8 and -11 are important and can induce non-canonical NLPR3 inflammasome activation in immune cells. Is there anything known in the kidney?

: We appreciate your comments.

1) Renal resident mononuclear cells (RMNCs), such as macrophages and dendritic cells, contain all parts of the NLRP3 inflamamsome (NLRP3, ASC, pro-caspase-1). Therefore, renal RMNCs can release mature proinflammatory cytokines and undergo caspase-1 dependent pyroptosis in response to LPS, silica, ATP, and radiocontrast (Kidney Int 2016;90:525-539, JCI 2013;123:236-246, JCI 2018;128:2894-2913). We add one sentence (revised line 67-69).

à NLRP3 inducers such as LPS, silica, ATP, and radiocontrast initiate cleavage of pro-caspase-1 and release of IL-1β and IL-18 in renal RMNCs or recruit leukocytes [8,15].

2) Not only renal RMNCs, but also renal parencymal cells (renal tubular epithelial cells (TECs), podocytes, glomerular endothelial cells, mesnagial cells) have a substantial amount of NLRP3 (Kidney Int 2015;87:74-84, Kidney Int 2018;93:656-669, Front Immunol 2018;9:2563, Cell death dis 2017;8:e2937). Podocytes secreted IL-18 and IL-1β in response to LPS + ATP or high glucose (J Pathol 2013;231:342-353, Hypertension 2012;60:154-162). Glomerular endothelial cells also cleaved pro-IL-1β ro active IL-1β under high glucose stimulation (Kidney Int 2015;87:74-84). In contrast, renal TECs did not express cleaved caspase-1 or release IL-1β in response to ATP, nigericin, monosodium urate or hypoxia (Cell Death Diff 2016;23:1331-1346, Front Immunol 2018;9:2563). Also, renal fibroblasts did not secrete IL-1β upon LPS + ATP (Kidney Int 2018;93:656-559). The contents of ‘1. NLRP3 in the kidney’ (revised line 70-91) were all revised in total.

à In addition to RMNCs, renal TECs, podocytes, glomerular endothelial cells, and mesangial cells contain a substantial amount of NLRP3[7,17-19]. Podocytes were identified as the primary source of IL-1β in glomerulosclerosis in both humans and mice [20,21]. Moreover, activated caspase-1 and IL-18 were increased in podocytes in response to LPS and ATP [22]. Homocysteine-induced NLRP3 inflammasome activation in murine podocytes caused cytoskeletal rearrangement in the podocytes [21]. In addition, high glucose-induced NLRP3 activation cleaved pro-IL-1β to active IL-1β in glomerular endothelial cells and podocytes [7].

Confusing results have been obtained regarding the expression of active caspase-1 and secretion of IL-1β in renal tubular epithelial cells (TECs). Albumin or oxidized LDL triggered the activation of the NLRP3 inflammasome and the secretion of IL-1β in TECs [23,24]. In contrast, we identified that various renal TECs, including HK-2, HKC-8, and RPTEC/TERT, and primary human renal proximal TECs did not express cleaved caspase-1 or release IL-1β under hypoxic conditions [18]. Similarly, transforming growth factor-beta (TGF-β)-stimulated renal TECs showed pro-IL-1β and very little IL-18 [12]. Another study also demonstrated that LPS combined with ATP, nigericin, monosodium urate crystals, or tumor necrosis factor (TNF)α/cycloheximide (CHX) failed to induce IL-1β or IL-18 release from mouse TECs [25]. Although this topic should be researched further for confirmation, we will introduce NLRP3 in renal TECs as an inflammasome-independent protein in this review.

Renal fibroblasts express NLRP3 but do not secrete IL-1β upon LPS and ATP treatment [17]. NLRP3 in renal fibroblasts was suggested to induce tubulointerstitial fibrosis via TGF-β signaling in an inflammasome-independent manner [17]. The data from mesangial cells are incomplete and controversial. Interestingly, mouse primary mesangial cells could not release IL-1β upon LPS and ATP stimulation; however, a rat mesangial cell line (HBZY-1) secreted IL-1β under high-glucose conditions, and human mesangial cells produced IL-1β after incubation with the IgA immune complex [26-29].

3) As you pointed out, NLRP3 can be activated without caspase-1. Macrophages secreted active IL-1β in response to LPS and cycloheximide depending on caspase-8 despite of absecne of caspase-1 (Cell death diff 2013;20:1149-1160). Also, LPS and oxidized phospholipides directly bind to caspase 11/4/5 (caspase-11 in mice, caspase-4 and caspase-5 in humans) and lead to the activatino of inflammasomes (Nature 2011;479:117-121, Eur J Immunol 2015;45:2911-2917). Although the activation of NLRP3 by these caspases have not been known in the kidneys, we added the contents in ‘2. Inflammasome-dependent NLRP3 in the kidney’ (revised lines of 117-124).

à Active caspase-1 also cleaves gasdermin-D (GSDMD), which induces pyroptotic cell death [39]. Generally, activation of the NLRP3 inflammasome via caspase-1 is defined as a canonical pathway. However, a recent study identified that the NLRP3 inflammasome could secrete IL-1β and induce pyroptosis without caspase-1 and GSDMD [40]. Macrophages secreted cleaved IL-1β under LPS and cycloheximide depending on caspase-8 during the absence of caspase-1and leaded to apoptotic cell death in response to nigericin [41,42]. Additionally, LPS and oxidized phospholipids directly bind to caspase 11/4/5 (caspase-11 in mice, caspase-4 and caspase-5 in humans) and lead to the activation of inflammasomes in macrophages without adaptor proteins [43,44]. This NLRP3 activation without ASC or caspase-1 are classified as the non-canonical inflammasome.

Figure 1: the figure legend is incomplete and difficult to follow. Please define the abbreviations e.g. IRI, LPS, ROS, PAMPs … and bring the terms in an alphabetic order.

: As your comments, we rewrite figure legend as following (revised line 92-102)

à NLRP3 in the kidney operates in an inflammasome-dependent or inflammasome-independent manner, depending on the DAMPs and stimulated cells involved. The NLRP3 inflammasome is activated in various renal diseases in renal resident mononuclear cells (RMNCs) and renal resident cells. Additionally, NLRP3 could regulate macrophage polarization via an inflammasome-independent way. NLRP3 led to the generation of mitochondrial ROS and tubulointerstitial fibrosis in renal TECs. NLRP3 in renal fibroblasts is implicated in mediating renal fibrosis. AIF = apoptosis-inducing factor, BAX = bcl-2-like protein 4, DAMPs = damage-associated molecular patterns, DC = dendritic cell, EC = endothelial cell, IgAN = IgA nephropathy, IRI = ischemic reperfusion injury, I-dependent = inflammasome-dependent, independent = inflammasome-independent, MФ = macrophage, MC = mesangial cell, NF-κB = nuclear factor kappa-light-chain-enhancer of activated B cells, ROS = reactive oxygen species, UUO = unilateral ureteral obstruction

Multiple references are missing: Line 64, 73, 107,111, 115, 130-134, 151, 202, 229, 243, 254, 256, 269, 270, 297, 307, 311, 318.

: All of the missing references were added.

Line 186-187: From my point of view this is an overstatement. Please rephrase the sentence.

: We edited the sentence as your recommendation (revised line 203-204).

à This suggested that NLRP3 in nonimmune renal cells could be critical for aggravating diabetic nephropathy.

In general, it is sometimes difficult for the reader to understand what models/cells/tissues species (human or mouse) is discussed for example see 192- 194.

: We edited the mansucript to clarify this issue (revised line 215-217).

à Dysregulation of endoplasmic reticulum stress and elevated fatty acids in obesity activate inflammatory signaling and result in insulin resistance. Ceramide activates the NLRP3 inflammasome in bone marrow-derived macrophages and adipose tissue.

Line 238-250: How about the inflammasome-independent mechanisms?

: So far, the role of inflammasome independent NLRP3 in lupus nephritis has not been clarified.

Minor comments:

Line 10-12: Please rephrase the sentence and include caspase-1 activation and cleavage of pro-IL-1beta and IL-18.

: We corrected the sentence (revised line 23-35).

à Cytoplasmic nucleotide-binding oligomerization domain-like receptor protein 3 (NLRP3) forms an inflammasome with ASC and pro-caspase-1, which is followed by the cleavage of pro-caspase-1 to active caspase-1 and ultimately the activation of IL-1β and IL-18 and induction of pyroptosis in immune cells.

Line 14-15: Please rephrase the sentence.

: We corrected the sentence (revised line 29-30).

à However, recent studies have suggested that NLRP3 has several inflammasome-independent functions in the kidney.

Line 20: ….in the kidney in both inflammasome-dependent….

: We corrected the sentence (revised line 32-34).

à This review will summarize the mechanisms by which NLRP3 functions in the kidney in both inflammasome-dependent and inflammasome-independent ways and the role of NLRP3 and NLRP3 inhibitors in kidney diseases.

Line 22: Please revise the Keywords.

: We corrected the sentence (revised line 36).

à Keywords: NLRP3; inflammasome; kidney; NLRP3 inhibitor

Line 52: a blank is missing.

: We corrected the sentence (revised line 56-58).

à Here, we will address what has been studied so far about the role of NLRP3, primarily focusing on the mechanism of action of the inflammasome-independent and inflammasome-dependent forms of NLRP3 in kidney diseases.

Line 56: define LPS

: LPS was defined (revised line 63).

à Lipopolysaccharide (LPS)

Line 164: are produced

: We modified the sentences in revised line 164 as a whole (revised 172-174).

à Renal tubular apoptosis and inflammation preceded renal leukocyte infiltration at 24 hours after glycerol-induced rhabdomyolysis [56]. Tubular injuries were clearly attenuated in NLRP3, ASC, caspase-1, and IL-1β KO mice during the early stage of rhabdomyolysis [56].

Line 243: imply

: We got rid of this sentence when we revised the manuscript.

Line 282: in a dose-dependent

: We insert ‘in a dose-dependent’ in the sentence (revised line 317-318).

à A recent study showed that tranilast inhibited the activation of caspase-1 and IL-1β release in macrophages in a dose-dependent manner [79].

Line 316: However, it…..

: Instead of ‘However, it’, we addeed ‘Notably’ in front of the sentence (revised line 349-350).

à Notably, dietary restriction or pharmacologic approaches to increase BHB could be a promising therapy for attenuating NLRP3-related inflammatory diseases.

Reviewer 3 Report

Some key findings describing the role of NLRP3 in kidney diseases were missing, as pointed out below. Mulay SR's review article published in Kidney Int (2019; 96: 58) has thoroughly discussed the contribution of the inflammasome-dependent and –independent functions of NLRP3 during the development and progression of chronic kidney disease in various experimental models and humans.  I could not see more new information from the current manuscript.

1. Some important papers discussing the role of mitochondrial oxidative stress and NLRP3 activation in kidney diseases are missing in this manuscript:
For example:
1) the role of the mitochondrial ROS-TXINP-NLRP3 biological axis in diabetic nephropathy (Redox Biology; 2018: 32;  Oxid Med Cell Longev. 2016; 2386068)
2)  NADPH oxidase-induced NALP3 inflammasome activation is driven by TXINP which contributes to podocyte injury in hyperglycemia (J Diabetes Res. 2015; 504761)
3)  Nod-like Receptor Protein 3 (NLRP3) Inflammasome Activation and Podocyte Injury via Thioredoxin-Interacting Protein (TXNIP) during Hyperhomocysteinemia. (JBC. 2014; 289: 27159)
4)  mROS-TXNIP axis activates NLRP3 inflammasome to mediate renal injury during ischemic AKI (Int J Biochem Cell Biol. 2018; 98: 43)
5) Activation of the TXNIP/NLRP3 inflammasome pathway contributes to inflammation in diabetic retinopathy: a novel inhibitory effect of minocycline (Inflamm Res. 2017; 66: 157)
2. Need to discuss the P2X7 receptor-NLRP3 axis in renal inflammation (J Pathol. 2013; 231: 342)
3. It has been recently reported that MCC950 ameliorates kidney injury in diabetic nephropathy by inhibiting NLRP3 inflammasome activation. (Diabetes Metab Syndr Obes; 2019: 12: 1297). The authors need to include discuss this article.
4. Line 43: need to add reference #27 here in addition to Ref 10-12.
5. Line 130-133 " similarly, we observed….". The authors need to add a reference to the statement described here.
6. Line 169-170: it is indicated here Data not shown. If the results were not published, the authors should not discuss them in the manuscript to support their conclusion.
7. Line 174-187: several observations were listed but not discussed well, making it difficult to read.
8. Line 194-196, the authors need to include Wen H et al. 's Nat Immunol paper (2011) when discussing the link between HF diet feeding, NLRP3 inflammasome, and adipose inflammation.

Author Response

Reviewer 3

Some key findings describing the role of NLRP3 in kidney diseases were missing, as pointed out below. Mulay SR's review article published in Kidney Int (2019; 96: 58) has thoroughly discussed the contribution of the inflammasome-dependent and –independent functions of NLRP3 during the development and progression of chronic kidney disease in various experimental models and humans. I could not see more new information from the current manuscript.

Some important papers discussing the role of mitochondrial oxidative stress and NLRP3 activation in kidney diseases are missing in this manuscript:

For example:

1) the role of the mitochondrial ROS-TXINP-NLRP3 biological axis in diabetic nephropathy (Redox Biology; 2018: 32;  Oxid Med Cell Longev. 2016; 2386068)

2)  NADPH oxidase-induced NALP3 inflammasome activation is driven by TXINP which contributes to podocyte injury in hyperglycemia (J Diabetes Res. 2015; 504761)

3)  Nod-like Receptor Protein 3 (NLRP3) Inflammasome Activation and Podocyte Injury via Thioredoxin-Interacting Protein (TXNIP) during Hyperhomocysteinemia. (JBC. 2014; 289: 27159)

4)  mROS-TXNIP axis activates NLRP3 inflammasome to mediate renal injury during ischemic AKI (Int J Biochem Cell Biol. 2018; 98: 43)

5) Activation of the TXNIP/NLRP3 inflammasome pathway contributes to inflammation in diabetic retinopathy: a novel inhibitory effect of minocycline (Inflamm Res. 2017; 66: 157)

: We appreciated your comments.

As you pointed out, mitochondrial disruption, ROS, and endogenous antioxidant inhibitor ‘TXNIP’ has been reported to associate with the activation of NLRP3 inflammasome (Nature Immunol 2013;14:454-460, Nature Immunol 2010; 11:136-140, JBC 2010;285:3997-4005). We added this concept (revised line 111-114).

à Thioredoxin-interacting protein (TXNIP), as an endogenous antagonist for thioredoxin, induces NLRP3 inflammasome activation in a ROS-sensitive manner [35]. It is speculated that TXNIP, translocated to mitochondria dysfunction under oxidative stress, cause NLRP3 inflammasome activation [36].

TXNIP inhibition attenuated diabetic nephropathy (Oxid Med Cell Longev 2016;2016:2386068, J Diabetes Res 2015;2015:504761). It was added (revised line 200-202)

à Inhibition of TXNIP abrogated diabetic nephropathy through inhibition of NLRP3 inflammasome activation in murine streptozotocin (STZ)-induced diabetes [64,65].

Need to discuss the P2X7 receptor-NLRP3 axis in renal inflammation (J Pathol. 2013; 231: 342)

: We appreciated your comments. The P2X7R is an important activator in NLRP3 inflammasome by extracellular ATP (Dev Comp Immunol 2012;38:312-320). The content was added (revised line 107-109)

à Three models have been suggested to progress NLRP3 inflammasome; K+ efflux by extracellular ATP through purinergic 2X7 receptor (P2X7R), reactive oxygen species (ROS) generation by PAMPs and DAMPs, and lysosomal disruption of the phagocytic vesicle including crystalline [31-33].

P2X7R is critical to drive high-fat diet-induced renal inflammation in podocyte (J Pathol 2013;231:342-353). We added the content (revised line 221-223). 

à P2X7R depletion protected mice from HFD-induced renal inflammation and fibrosis [22]. The authors showed that NLRP3 inflammasome activation was attenuated in P2X7R-silenced podocytes under LPS with ATP stimulation.

It has been recently reported that MCC950 ameliorates kidney injury in diabetic nephropathy by inhibiting NLRP3 inflammasome activation. (Diabetes Metab Syndr Obes; 2019: 12: 1297). The authors need to include discuss this article.

: We added related content and reference (revised line 361-362).

à Intraperitoneal MCC950 treatment in db/db mice decreased albuminuria, serum creatinine, and pathologic changes, and improved renal cortical fibrosis [29].

Line 43: need to add reference #27 here in addition to Ref 10-12.

: We added the reference (Mol Cell Endocrinol 2018;478:115-125).

Line 130-133 " similarly, we observed….". The authors need to add a reference to the statement described here.

: We added the reference (Front Immunol 2018;9:2563).

Line 169-170: it is indicated here Data not shown. If the results were not published, the authors should not discuss them in the manuscript to support their conclusion.

: We agree with your opinion. We deleted the results not published.

Line 174-187: several observations were listed but not discussed well, making it difficult to read.

: We edited the paragraph easily to understand (revised line 194-213).

à Hyperglycemia and its related metabolic rearrangement could act as DAMPs that are detected by NLR receptors [61,62]. The transcripts of NLRP3, ASC, and proinflammatory cytokines increased in monocytes from patients with type 2 DM [63]. Intriguingly, an increase in plasma levels of IL-1β and IL-18 occurred prior to the presentation of albuminuria and renal histologic changes in murine diabetic nephropathy [7]. Therefore, systemic activation of the NLRP3 inflammasome might play an essential role in the occurrence and progression of diabetic nephropathy. NLRP3 KO mice demonstrated repression of diabetic nephropathy in type 1 and type 2 diabetes by blocking NLRP3-mediated mitochondrial ROS generation [7,10]. Inhibition of TXNIP abrogated diabetic nephropathy through inhibition of NLRP3 inflammasome activation in murine streptozotocin (STZ)-induced diabetes [64,65]. However, diabetic nephropathy was improved in WT bone marrow transplanted-NLRP3 KO mice but not in NLRP3 KO bone marrow transplanted-WT mice [7]. This suggested that NLRP3 in nonimmune renal cells could be critical for aggravating diabetic nephropathy. Podocytes and glomerular endothelial cells contain NLRP3 and release cleaved IL-1β under high-glucose stimulation. Additionally, high glucose led to NLRP3 inflammasome activation in mesangial cells, and NLRP3 inhibition decreased IL-1β release from mesangial cells [19,29,65]. NLRP3-silenced renal TECs were also protected from high glucose-induced ROS injury [10]. We demonstrated that uric acid-stimulated renal TECs recruited macrophages via C-X-C motif chemokine 12 and high mobility group box 1 (HMGB1), and then the macrophages produced IL-1β, which again resulted in NF-κB activation in TECs [66]. The cross-talk between macrophages and renal TECs contributed to boosting inflammation in the kidney in OLETF rats [66]. Overall, NLRP3 plays a pivotal role in diabetic kidney disease progression

Line 194-196, the authors need to include Wen H et al. 's Nat Immunol paper (2011) when discussing the link between HF diet feeding, NLRP3 inflammasome, and adipose inflammation

: We added the reference.

Round 2

Reviewer 1 Report

The authors have addressed all of my concerns.

Reviewer 2 Report

The review of Kim et al. is nicely summarizing the mechanism of NLRP3 inflammasomes in the kidney. The focus of this review is on both the inflammasome-dependent and inflammasome-independent mechanisms as well as the role of NLRP3 and NLRP3 inhibitors in kidney diseases.

Reviewer 3 Report

The manuscript was significantly improved.

Minor comments:

Line 105: it should be "NF-κB activators". Please correct the typo.

Line 327-335: may want to rewrite these sentences to summarize the effects of Bay11 on NLRP3 inflammasome activation.